# DC3 is a method for deconvolution and coupled clustering from bulk and single-cell genomics data

Wanwen Zeng[1,2,5], Xi Chen[1,5], Zhana Duren[1,5], Yong Wang[3,4], Rui Jiang[2]* & Wing Hung Wong[1]*

Characterizing and interpreting heterogeneous mixtures at the cellular level is a critical problem in genomics. Single-cell assays offer an opportunity to resolve cellular level heterogeneity, e.g., scRNA-seq enables single-cell expression profiling, and scATAC-seq identifies active regulatory elements. Furthermore, while scHi-C can measure the chromatin contacts (i.e., loops) between active regulatory elements to target genes in single cells, bulk HiChIP can measure such contacts in a higher resolution. In this work, we introduce DC3 (De-Convolution and Coupled-Clustering) as a method for the joint analysis of various bulk and single-cell data such as HiChIP, RNA-seq and ATAC-seq from the same heterogeneous cell population. DC3 can simultaneously identify distinct subpopulations, assign single cells to the subpopulations (i.e., clustering) and de-convolve the bulk data into subpopulation-specific data. The subpopulation-specific profiles of gene expression, chromatin accessibility and enhancer-promoter contact obtained by DC3 provide a comprehensive characterization of the gene regulatory system in each subpopulation.

[1] Department of Statistics, Department of Biomedical Data Science, Bio-X Program, Stanford University, Stanford, CA 94305, USA. [2] MOE Key Laboratory of Bioinformatics, Bioinformatics Division, Department of Automation, Beijing National Research Center for Information Science and Technology, Tsinghua University, Beijing 100084, China. [3] CEMS, NCMIS, MDIS, Academy of Mathematics and Systems Science, Chinese Academy of Sciences, Beijing 100080, China. [4] Center for Excellence in Animal Evolution and Genetics, Chinese Academy of Sciences, Kunming 650223, China. [5] These authors contributed equally: Wanwen Zeng, Xi Chen, Zhana Duren *email: ruijiang@tsinghua.edu.cn; whwong@stanford.edu

With the rapid development of single-cell (sc) genomics technology, researchers are now able to study heterogeneous mixtures of cell populations at the single cell level. Each type of sc-genomics experiments offers one particular aspect to delineate the heterogeneity; for example, scRNA-seq[1] enables single cell gene expression profiling, scATAC-seq[2] identifies accessible chromatin regions in single cells and scHi-C[3] measures chromatin contacts in the single-cell level. In many situations, a first step in the analysis of single-cell data is clustering, that is, to classify cells into the constituent subpopulations. While clustering methods for scRNA-seq or scATAC-seq alone have been widely studied[4, 5], when different types of sc-genomics experiments are performed on different samples from the same heterogeneous cell population, then all samples are informative on the underlying subpopulations, and analysis of one sample should be informed by the analysis on another sample. Recently, Duren et al.[6] proposed a coupled NMF (coupled non-negative matrix factorization) method to cluster cells in scRNA-seq and scATAC-seq samples and to infer both the expression profile and accessibility profile of each subpopulation. These two profiles reveal a great deal about the subpopulation of cells: the accessible regions identify the active regulatory elements (RE) while the expression profiles identify actively transcribed genes[7]. However, even with these two profiles, our understanding of the subpopulation-specific regulatory networks remains incomplete if we cannot link the active REs to their target genes. In principle, such linkages can be obtained by measuring 3D contacts between REs and gene promoters. In bulk sample, it is easy to measure 3D contacts between active enhancers and gene promoters by H3K27ac HiChIP experiments[8]. On the other hand, combinatorial indexing can be used for 3D contact measurement in single cells[3].

In order to take these 3D contacts into account in the study of subpopulation-specific regulatory networks, here we introduce DC3 as a method for the joint analysis of bulk and single cell data under various settings of input data combinations, including: (1) scRNA-seq, scATAC-seq and scHi-C; (2) scRNA-seq, scATAC-seq and bulk HiChIP; (3) scRNA-seq, bulk ATAC-seq, bulk HiChIP; (4) bulk RNA-seq, scATAC-seq, bulk HiChIP. Based on comprehensive simulation experiments, we show that this method can deconvolve bulk profiles into subpopulation-specific profiles. At the same time, the subpopulation-specific profiles in turn leads to improved coupled clustering results of single-cell data. To assess its performance in a heterogeneous cell population in vivo, we apply DC3 on a population obtained after four days of retinoic-acid (RA) induced differentiation of mouse embryonic stem cells. We validated the HiChIP profile for one of the inferred subpopulations by showing its consistency to HiChIP data on cells obtained by fluorescence-activated cell sorting (FACS) based markers specific to that subpopulation. Finally, we illustrate the value of results from DC3 by using them to derive the core regulatory network and their downstream effectors in each of the subpopulation in the induced differentiated mouse embryonic stem cells.

## Results

**The DC3 algorithm**. We formulate the joint analysis of bulk and single cell RNA-seq, ATAC-seq and Hi-C data as an optimization problem (Methods). For each type of single cell data, the cost function contains a NMF term that drives clustering of the single cells through non-negative matrix factorization (NMF). For each bulk data type, the cost function contains a coupling term that couples the three data types within each subpopulation by enforcing certain relationship among them. For example, suppose we have an input data setting with scRNA-seq, scATAC-seq and bulk HiChIP (Fig. 1a), then the cost function is given in Fig. 1b

where the first term gives the coupling and the other two are NMF terms. As previously described[6], each NMF term drives the decomposition of a single-cell data matrix into two factors **W** and **H**, with columns of **W** representing cluster-specific profiles, and each column of H giving the relative weights (for cluster-assignment) of a particular single cell. To derive the coupling, we examined data from various cell lines and found that HiChIP loop counts are generally positively correlated with both gene expression values from RNA-seq (Supplementary Fig. 1) and enhancer openness from ATAC-seq (Supplementary Fig. 2). This observation motivated us to use a linear relation between the loop count and the product of gene expression and enhancer openness to couple the three data types, which gives rise to first term of the cost function. This approach can be extended to handle any combination of bulk and single cell data, as long as at least one of the data types contains single cell data. The general cost function and further discussions are given in the Methods section. Note that, instead of using a pre-defining enhancer set, DC3 defines the candidate enhancers directly based on ATAC-seq and HiChIP data.

The main purpose of the coupling term is to improve clustering of single cells by exploiting the statistical correlation between different data types within each subpopulation. Although the optimization can provide estimates of subpopulation-specific profiles (subpopulation profiles) using the **W** matrix in the NMF term, in simulation experiments we observed that when single cell data is available, we can better estimate a subpopulation profile by averaging the data from the single cells assigned to that subpopulation (Methods and Supplementary Table 1). For a data type with only bulk data, we can obtain its subpopulation profiles by a simple method using an expression based on the already estimated profiles of the single cell data types (Methods). Alternatively, we may infer its subpopulation profiles based on a Poisson model with the profiles of the single cell data types treated as known (Methods). Compared to the simple method, the Poisson model-based method has a better interpretation and also performs slightly better in simulations (Supplementary Fig. 3 and Supplementary Table 2), but at the cost of a much higher time complexity (24 h vs 3 min). For computational efficiency, DC3 uses the simple method as default.

**Evaluation of deconvolution on in silico mixture of cells**. We used deconvolution to denote the task of estimating subpopulation profiles regardless of whether it was based on single cell or bulk data. We constructed an in silico mixture of deep single cell data (UMI~I million per cell) from two cell lines, GM12878 and K562 (Methods) and used it to evaluate the performance of our method under four settings of input data combinations: (1) scRNA-seq, scATAC-seq and scHi-C; (2) scRNA-seq, scATAC-seq and bulk HiChIP; (3) scRNA-seq, bulk ATAC-seq, bulk HiChIP; (4) bulk RNA-seq, scATAC-seq, bulk HiChIP. Deconvolution performance was assessed by the mean Pearson correlation coefficient (mean PCC scores in 50 runs of DC3) between the observed versus predicted subpopulation-specific profiles in the two cell lines (Methods). DC3 was seen to perform well in HiChIP, RNA-seq and ATAC-seq deconvolution, with mean PCC score of 0.78–0.95 in HiChIP deconvolution, 0.85–0.99 in RNA-seq deconvolution, 0.88–0.99 in ATAC-seq deconvolution (Table 1). As a comparison, we generated a null distribution by randomly assigning the reads to two artificial cell lines and repeated the whole computation (Table 1 and Supplementary Figs. 4–9). The deconvolution accuracy of DC3 was seen to be significantly higher than random deconvolution.

To assess the impact of sequencing depths, we further conducted a series of dropout experiments, where site-level dropout in both scRNA-seq, scATAC-seq, and scHi-C data were

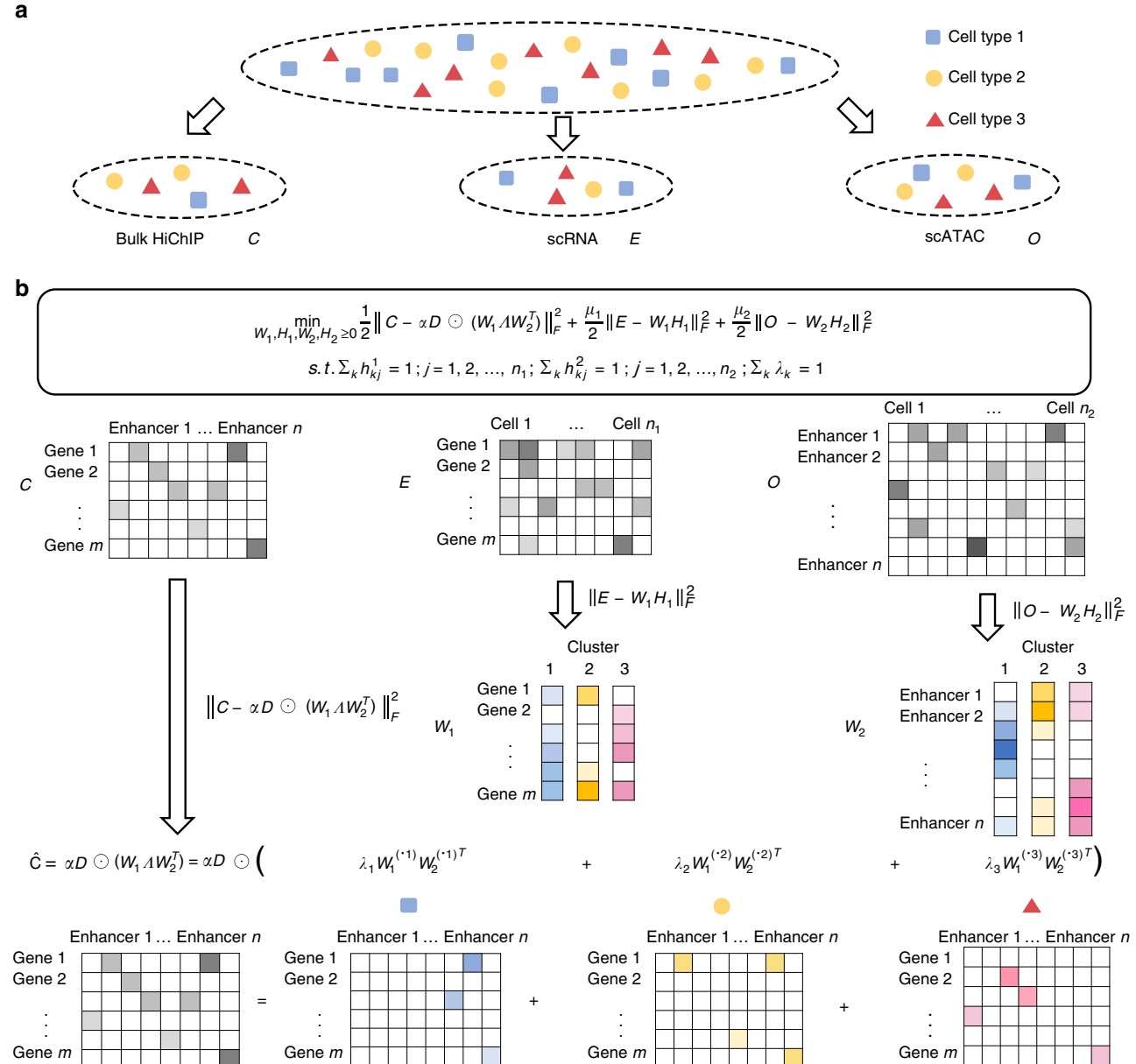

**Fig. 1** Overview of the DC3 method. **a** DC3 performs joint analysis using three types of data from separate samples from the same cell population: scRNA-seq, scATAC-seq, bulk HiChIP. $E$ denotes the genes expression level in each cell measured in scRNA-seq; $O$ denotes enhancer chromatin accessibilities in each cell measured in scATAC-seq; $C$ denotes the enhancer-promoter interactions strength (loop counts) between each gene and each enhancer measured in bulk HiChIP. **b** A graphical example for simultaneously decomposing $E$, $O$, $C$ to get the underlying clusters and cluster-specific HiChIP in $K = 3$ case: (1) $\|E - W_1 H_1\|_F^2$: $w_{ik}^1$ gives the mean gene expression for the $i$-th gene in the the $k$-th cluster of cells, while $h_{kj}^1$ gives the assignment weights of the $j$-th cell to the $k$-th cluster; (2) $\|O - W_2 H_2\|_F^2$: $w_{ik}^2$ gives the mean chromatin accessibility for the $i$-th enhancer in the $k$-th cluster of cells, while the $j$-th column of $H_2$ gives the assignment weights of the $j$-th cell to the different clusters; (3) $\hat{C} = \alpha D \odot (W_1 \Lambda W_2^T)$: each enhancer–promoter interaction $c_{ij}$ can be decomposed into subpopulation-specific interactions, i.e. $c_{ij} = \lambda_k \sum_k c_{ijk}$, where $c_{ijk}$ is the interaction strength in the $k$-th subpopulation and $\lambda_k$ is proportional to the size of the subpopulation; $\Lambda$ is a $K$ by $K$ diagonal matrix $[\lambda_1, \lambda_2, \dots \lambda_K]$. Within each subpopulation, following the assumption that an enhancer-promoter interaction is proportional to the product of accessibilities of the corresponding enhancer and promoter, we model $c_{ij}$ as $c_{ij} = \alpha d_{ij} \sum_k \lambda_k w_{ik}^1 w_{jk}^2$, where $d_{ij}$ is a set of indicators selecting the enhancer-promoter pair to be modeled. Therefore, cluster-specific HiChIP interactions of $k$-th subpopulation can be obtained from the $k$-th column of $W_1$ multiple the transposition the $k$-th column of $W_2$ : $\alpha D \odot \left( \lambda_k W_1^{(\cdot k)} W_2^{(\cdot k)T} \right)$

simulated using different dropout rates (Methods). The results were presented in Supplementary Tables 3–5. As expected, for all four input settings, deconvolution accuracy deteriorated with increasing dropout rate. When only one data type is available in single cells (input settings 3 and 4) and when the dropout rate is high, we cannot obtain significantly better performance over random deconvolution. On the other hand, when both RNA-seq and ATAC-seq were available in single cells (input settings 1 and 2), DC3 deconvolution performance was still acceptable (PCC 0.82–0.93) at 80% dropout and remained significantly better than

**Table 1 The deconvolution performance of DC3 with different input combinations**

| Input combinations | HiChIP | | RNA-seq | | ATAC-seq | | Mean |
|---|---|---|---|---|---|---|---|
| | K562 | GM12878 | K562 | GM12878 | K562 | GM12878 | |
| scRNA-seq, scATAC-seq and scHi-C | 0.92 ± 0.01 | 0.95 ± 0.01 | 0.98 ± 0.00 | 0.99 ± 0.00 | 0.98 ± 0.01 | 0.99 ± 0.01 | 0.97 |
| scRNA-seq, scATAC-seq and bulk Hi-C | 0.86 ± 0.00 | 0.95 ± 0.00 | 0.98 ± 0.00 | 0.99 ± 0.00 | 0.98 ± 0.01 | 0.99 ± 0.01 | 0.95 |
| scRNA-seq, bulk ATAC-seq, bulk Hi-C | 0.78 ± 0.09 | 0.85 ± 0.08 | 0.95 ± 0.04 | 0.94 ± 0.02 | 0.85 ± 0.02 | 0.85 ± 0.02 | 0.87 |
| Bulk RNA-seq, scATAC-seq, bulk Hi-C | 0.71 ± 0.09 | 0.83 ± 0.08 | 0.85 ± 0.02 | 0.88 ± 0.03 | 0.88 ± 0.01 | 0.89 ± 0.01 | 0.84 |
| Random deconvolution | 0.61 ± 0.12 | 0.76 ± 0.10 | 0.76 ± 0.11 | 0.74 ± 0.08 | 0.62 ± 0.12 | 0.71 ± 0.08 | 0.70 |

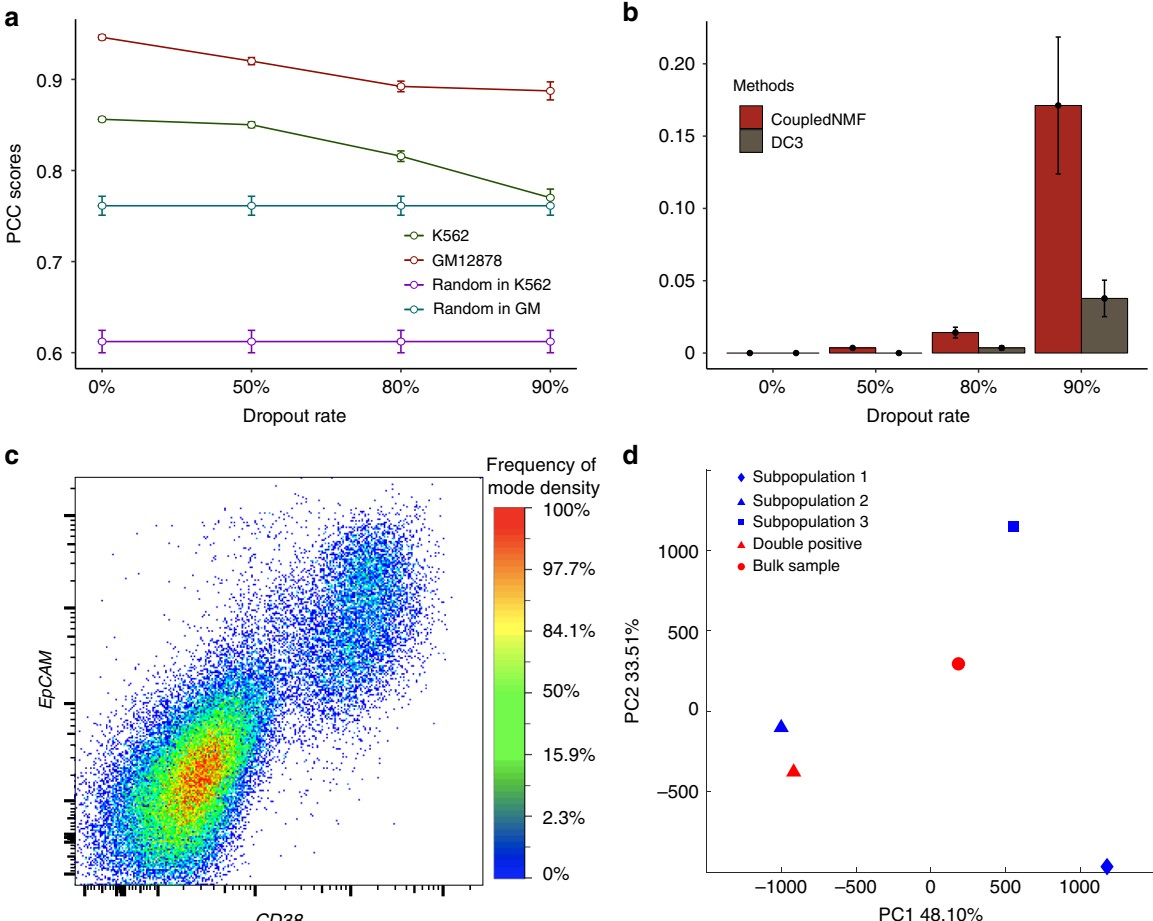

**Fig. 2** Validation of DC3 in simulation and real data. **a** Performance of HiChIP deconvolution scRNA-seq and scATAC-seq for GM12878 and K562 simulation data under different drop out rates. As a comparison, the random deconvolution results are presented. Error bars represent standard deviation. **b** Performance of joint clustering for GM12878 and K562 simulation data under different drop out rates. Error bars represent standard deviation. **c** FACS plot shows that in RA-day 4, 15.7 ± 3.2% cells are double positive for population-2-markers *EpCAM* and *CD38*. **d** Performance of HiChIP deconvolution in RA-day 4 real data. The HiChIP profile measured from double positive cells (red triangle) is much closer to that inferred for subpopulation 2 (blue triangle) than to the HiChIP profiles inferred for the other subpopulation (blue circle and blue rhombus) or measured from the bulk sample (red circle). All HiChIP profiles are represented using *n*-dimensional vectors with each dimension indicates corresponding loop counts. Source data are provided as a Source Data file

random deconvolution even at 90% dropout. It is noteworthy that at all dropout levels (Fig. 2a), deconvolution accuracy in setting 2 (scRNA, scATAC, bulk HiChIP) was comparable to that in setting 1 (scRNA, scATAC, scHiChIP). In the remainder of this paper, we will focus on the further evaluation and application of DC3 under setting 2.

**Evaluation of clustering on in silico mixture of cells**. In this and subsequent sections, we assume that the input data setting is scRNA-seq, scATAC-seq and bulk HiChIP. First, we investigated

whether DC3 would lead to improved clustering performance as compared to clustering without the incorporation of loop data from HiChIP. We compared DC3 to coupled NMF (which performed coupled clustering without using the loop data) and NMF (which performed clustering separately for each single-cell data type) in in silico mixture of GM12878 and K562 cells. Fifty independent runs were performed for each method. At each dropout rate, we compared clustering results based on the average (over the scRNA-seq and scATAC-seq samples) error rate in cluster assignment. Figure 2b shows the results for DC3 and coupled NMF (details including NMF results in Supplementary

**Table 2 Subpopulation-specific GO terms enrichment results**

| -log10(*p*-value) | | Original | | Down-sampling | |
|---|---|---|---|---|---|
| | | scRNA-seq | scRNA-seq + HiChIP | scRNA-seq | scRNA-seq + HiChIP |
| Subpop 1 | neuron development | 37.83 | 37.84 | 16.81 | 24.26 |
| | axon development | 26.44 | 27.63 | 11.52 | 17.87 |
| | axonogenesis | 18.12 | 20.12 | 10.01 | 16.55 |
| | neuron projection guidance | 16.73 | 17.88 | 7.99 | 16.03 |
| Subpop 2 | cardiovascular system development | 9.13 | 11.34 | 4.87 | 11.02 |
| | vasculature development | 9.76 | 11.75 | 5.59 | 8.58 |
| | circulatory system development | 7.17 | 9.93 | 3.89 | 8.87 |
| | muscle structure development | 5.98 | 8.72 | 2.88 | 7.33 |
| Subpop 3 | forebrain development | 11.32 | 13.26 | 1.44 | 13.03 |
| | central nervous system development | 10.74 | 14.34 | 1.40 | 11.96 |
| | brain development | 9.02 | 13.98 | 2.18 | 11.22 |
| | head development | 8.77 | 12.05 | 1.57 | 9.51 |

The enrichment *p*-values are transformed to −log10(*p*-values) and shown in the table. Original: scRNA-seq measured in SMART-seq with median ~1 million reads per cell; Down-sampling: simulated scRNA-seq measured in Drop-seq with median UMI ~5000

Table 6). Since the two cell types were rather distinct, with the initial deeply sequenced mixtures (similar in depth as data from Fluidigm), both methods performed well with no cells misclassified in any run. As the dropout rate increased, the two types of cells became less distinct (Supplementary Figs. 10–13) and the incorporation of loop data became more important. In particular, the incorporation of loop data reduced the classification error by more than four folds when the dropout rate is at 80% or higher, which corresponds to a sequencing depth typical of data from droplet-based system such as 10× (median UMI < 10,000). As a comparison, we performed DC3 in three negative control experiments, by keeping two of the three data sets the same and randomly permutating the third data set (Supplementary Tables 7–9). The performance of DC3 dropped in these negative controls, indicating that each data type was important to clustering results. These results demonstrated the potential of DC3 to improve clustering of single cells.

**Evaluation of deconvolution on experimental mixtures**. When mouse embryonic stem cells (mESC) are induced to differentiate, several different lineages of cells may emerge, resulting in a real experimental mixture suitable for analysis with our approach. Specifically, embryoid bodies (EBs) are obtained from mESCs using the hanging drop method and then differentiated further under retinoic acid (RA) treatment (Methods). We performed scRNA-seq, scATAC-seq and bulk HiChIP on the mixture after 4 days of RA treatment (RA-day 4). The scRNA-seq and scATAC-seq samples have already been analyzed in our previous study[6]. We wanted to assess the performance of DC3 in the joint analysis of the three data types together. DC3 identified 3 subpopulations (Supplementary Figs. 14, 15) together with their subpopulation-specific loop profiles. Previous study has shown that subpopulation 1 and subpopulation 3 were two related subpopulations[6]. To isolate pure cell population, we focused on the more distinct subpopulation (subpopulation 2) and performed further experiments to validate its inferred loop profile. To this end, we searched for subpopulation 2-specific surface markers and identified EpCAM and CD38 as being highly expressed in cluster 2 but not in subpopulations 1 and 3 (Methods, Supplementary Figs. 16, 17). We performed FACS experiments by using these two markers to isolate subpopulation 2 cells. Figure 2c shows that we successfully isolated 15.7 ± 3.2% EpCAM/CD38 double positive cells at RA-day 4 (Supplementary Fig. 18).

Next, we performed a HiChIP experiment to obtain the loop profile for these double positive cells. We noted that 5 independent FACS runs were necessary in order to collect enough cells for HiChIP. From the PCA plot of the loop profiles (Fig. 2d), we can see that the double positive sample was indeed far closer to subpopulation 2 than subpopulation 1, 3, or the bulk sample. The loop profile of double positive cells had a PCC of 0.7633 with that of subpopulation 2 cells (Supplementary Fig. 19), which was far higher than its PCCs with the profiles of the other two subpopulations (0.34 and 0.45; Supplementary Figs. 20, 21). Together, these results validated the performance of DC3 in the deconvolution of loop data in a real biological mixture.

**DC3 improves interpretation of subpopulations**. To assess whether adding loop data will help to interpret the top genes in each subpopulation, we carry out GO terms enrichment analysis. First, we combined the scRNA-seq data and the loop profile for each subpopulation to select the top 1000 subpopulation-specific genes, and performed gene ontology (GO) terms enrichment analysis (Methods). We compared the enrichment results to those obtained when the selection of subpopulation-specific genes was based on scRNA-seq data alone. Table 2 (under original) gives the most enriched GO terms and *p*-values in each subpopulation. Although both subpopulations 1 and 3 are strongly enriched in nervous system-associated terms, subpopulation 1 is specific to neuron development and contains terms like axonogenesis and neuron projection guidance. Meanwhile, subpopulation 3 is enriched in terms concerning general brain and central nervous system development. Finally, subpopulation 2 has weaker but still highly significant enrichments in mesodermal development terms such as muscle structure development and cardiovascular development.

We further investigate how the enrichment results depend on sequencing depths by a down-sampling experiment (Methods), and the results are shown in Table 2 (under Down-sampling). Consistent with previous benchmark studies[9], deep scRNA-seq allows much better characterization (in the sense of high enrichment scores) of the subpopulations than the low-depth scRNA-seq. On the other hand, as shown by the −log₁₀(*p-value*), adding loop profile information offers an improved characterization of the subpopulations over scRNA-seq alone at any levels of sequencing depth. The improvement is especially large when the sequencing depth is low. In conclusion, DC3-inferred loop information can be used to improve the interpretation of subpopulations of cells.

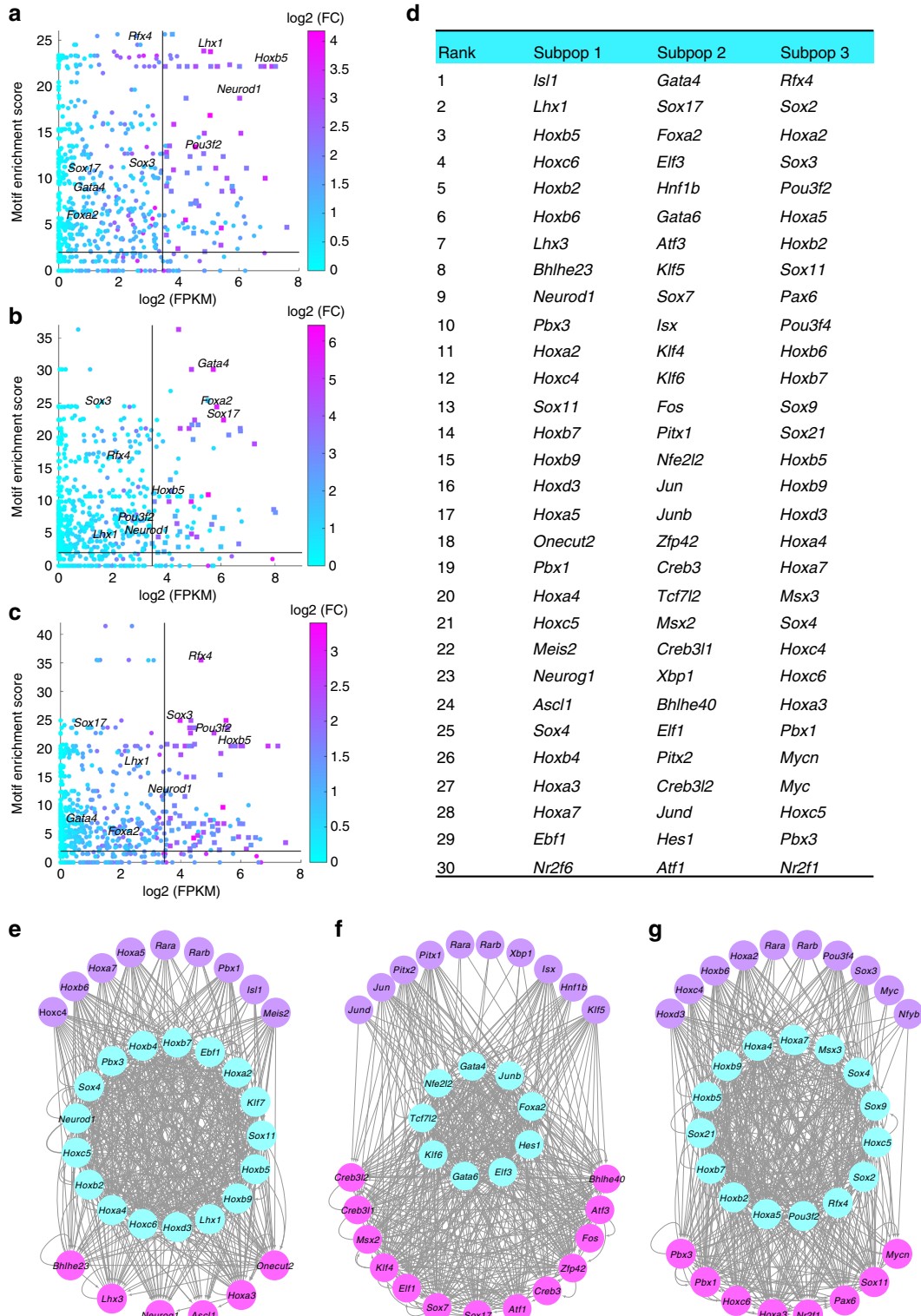

**Fig. 3** Analysis of subpopulation-specific regulatory networks. **a–c** Scatter plots of TF expression level and motif enrichment scores in the three subpopulations in RA-day 4. Node color represents expression specificity. Horizontal and vertical black lines indicate threshold values of motif enrichment scores and TF expression level. Key TFs are represented by squares (see text for key TF definition). **d** Top 30 key TFs in each subpopulation. Ranking is based on the product of log2(FPKM), motif enrichment score and expression specificity. **e–g** Dense subnetworks of key TFs plus expressed RA receptors in subpopulations 1 to 3 (left to right). Cadet blue color nodes represent the core subnetwork, violet nodes represent the upstream subnetwork and pink nodes represent the downstream subnetwork. Only the top 30 key TFs are shown. Source data are provided as a Source Data file

**Subpopulation accessibility, expression, and loop profiles.** The chromatin accessibility, gene expression and 3D contact profiles provided by DC3 for each subpopulation can be used to construct subpopulation-specific gene regulatory networks. However, typically these networks are large and complex, which makes it difficult to discern the key regulatory relationships. Therefore, we have developed a method to extract and visualize important subnetworks. Below, we demonstrated this method in the RA-day 4 example.

(Step 1) Identification of key regulators for each subpopulation: We merged the scRNA-seq reads from cells in the same subpopulation and calculated subpopulation-specific gene expression (in FPKM). Similarly, we merged the scATAC-seq cells in each subpopulation to call open peaks and computed motif enrichment scores (Methods). We defined the key regulators of a subpopulation as those TFs with high expression level (FPKM > 10), high motif enrichment score (>2), and differential expression compared to at least one of the other subpopulations (t-test, adjusted p-value <0.01). There are 58, 42, and 71 key regulators for subpopulations 1, 2, and 3 respectively (Fig. 3a–c, Supplementary Table 10–12). We ranked the key TFs by its importance score, defined as the product of its expression (i.e., log2 of FPKM), expression specificity (i.e., maximum expression fold change compared to the other two subpopulations) and motif enrichment score. The Top 30 key regulators are shown in Fig. 3d. For example, {*Lhx1*, *Neurod1*}, {*Gata4*, *Sox17*, *Foxa2*} and {*Rfx4*, *Sox3*} are high ranking specifically in subpopulations 1, 2 and 3 respectively, while *Pou3f2* and several *Hox* genes are high ranking in both subpopulations 1 and 3.

(Step 2) Construction of gene regulatory networks: On each subpopulation, we identified enhancer-target gene pairs with loop counts greater than or equal to 2. Given an enhancer-target gene pair, we connect it to key TFs which have both significant motif match on the enhancer region and significant correlation with target gene in the single cell gene expression data. This gives 14,979, 4,909 and 15,459 TF-Enhancer-Gene triplets in subpopulations 1, 2, and 3 respectively. Finally, for any pair of TF and target gene, say $T_i$ and $G_j$, we compute a TF-Target score $W_{ij}$ as the sum, over TF-RE-Gene triples with TF = $T_i$ and Gene = $G_j$, of the product of the motif score of $T_i$ on the RE and the loop count between RE and $G_j$. In this way, we obtained a regulatory networks for each subpopulation, defined as the directed graph with key TFs are nodes and TF-Target scores[10] as edge weights. The networks for the three subpopulations contain (58 nodes, 1043 edges), (42 nodes, 685 edges) and (71 nodes, 1037 edges) respectively.

(Step 3) Analysis of dense subnetwork: For each subpopulation-specific network, we extracted its dense subnetwork by quadratic programming (Methods). The extracted subnetwork is seen to be significantly denser than those obtained from random networks with same in-degree and out-degree for each node as our network (p-value equals 0.0230, 0.0180 and 0.0320 in subpopulations 1, 2, and 3, see Methods). The dense subnetwork was further partitioned into (i) the core subnetwork consisting of TFs that densely cross-regulate each other to achieve robust maintenance of the cellular state, (ii) the upstream subnetwork consisting of TFs that may regulate the core, and (iii) the downstream subnetwork consisting of key TFs regulated by the core. Different downstream TFs may be involved in different pathways or functions characteristic of the cells in the subpopulation. Figure 3e–g present the dense subnetworks of the three subpopulations. Downstream TFs in subpopulation 1 included *Ascl1*, *Neurog1*, *Lhx3*, *Onecut2* and *Bhlhe23*. The BHLH transcription factor *Ascl1* is one of the most important factors in neural commitment and differentiation[11], and it is also necessary for reprograming from fibroblasts to functional neurons[12]. *Lhx3* in known to contributes to the specification of motor neuron[13]. In

subpopulation 2, *Foxa2*, *Gata4*, and *Gata6* are in the core subnetwork. *Foxa2* is a pioneer factor important in mesendoderm development and is known to regulate *Gata4*[14], *Gata4* and *Gata6* are master TFs important to heart and gut formation. Our analysis suggests that these core TFs, together with their downstream effectors such as *Sox17*, may drive differentiation towards mesodermal and endoderm lineages. In subpopulation 3, *Rfx4 and Pou3f2* are in the core subnetwork. A novel splice variant of *Rfx4* is reported to be crucial for normal brain development[15] and *Pou3f2* is involved in cognitive function as well as adult hippocampal neurogenesis[16]. Downstream TFs in subpopulation 3 included *Pax6*. *Pax6* is important for the maintenance of brain integrity[17]. We note that many *Hox* genes are found in the core subnetworks of subpopulations 1 and 3, suggesting that they are important in the maintenance of these neural related populations. On the other hand, *Lhx1*[18] and *Neurod1*[19] are specific to subpopulation 1 while *Rfx4*[20] and *Pax6*[21] are specific to subpopulation 3. These regulators may play a role in defining the differences of these two related but distinct subpopulations.

## Discussion

In summary, we developed DC3 for simultaneous deconvolution and coupled clustering based on the joint analysis of different combinations of bulk and single-cell level RNA-seq, ATAC-seq, and HiChIP data. We showed that DC3 can decompose bulk profiles into subpopulation-specific profiles and at the same time enhance clustering performance of the single-cell data. The subpopulation-specific HiChIP interactions are seen to lead to improved interpretation of the subpopulations. Furthermore, we showed that the accessibility, expression and loop profiles inferred by DC3 can serve as a foundation for further analyses of the regulatory systems, such as the extraction of core subnetworks, in a population-specific manner.

Since DC3 is an unsupervised method and the hyper-parameters can be tuned automatically, it can be applied to many different scenarios. For example, existing single cell atlases[22–25] usually adopt barcode-based Drop-seq experiments (median UMI ~5000). If scRNA-seq and scATAC-seq with low sequencing depths have already been performed in the same cell population, then with additional simple HiChIP experiments, DC3 can greatly improve the characterization of the different subpopulations and their regulatory networks. As another example, if scRNA-seq, scATAC-seq and bulk HiChIP are performed in heterogeneous tumor cell population, DC3 can help to distinguish the subpopulations in the mixture and identify the TFs, enhancers and genes that are important in the subpopulations.

Finally, our optimization-framework is flexible and can be extended easily. For example, recently Cao et al. proposed a combinatorial indexing–based assay sci-CAR that jointly profiles chromatin accessibility and mRNA in each of thousands of single cells[26]; Lin et al.[27] proposed a model-based method to infer the subpopulations. In the future we will modify the cost function of DC3 to incorporate data types from such emerging single cell experiments, and incorporate the model-based method into the inference.

| Table 3 The numbers of cells in simulation data | | | |
|---|---|---|---|
| | **GM12878** | **K562** | **Total** |
| scRNA-seq | 73 | 73 | 146 |
| scATAC-seq | 373 | 373 | 746 |
| scHi-C | 100 | 100 | 200 |

## Methods

**Data preprocessing**. We aligned scATAC-seq reads to reference genome mm9 and removed duplicates. MACS2[28] was employed to call peaks by merging reads from all single cells and removed peaks present less than 10 cells. The final read counts for each peak on each cell were calculated by bedtools[29] intersect command.

We mapped scRNA-seq reads to mm9 by STAR[30] following ENCODE[31] pipeline and calculated Transcripts Per Million (TPM) by RSEM[32] using GENCODE[33] vM16 annotation.

We used HiC-pro[34] to process HiChIP data from raw fastq files to normalized contact maps using reference genome mm9. Then hichipper[35] was employed to perform bias-corrected peak calling, library quality control and DNA loop calling. We filter out the replicates that have less than 500 strong loops (greater than 5 reads). We further utilized ATAC-seq peaks to annotate loops and to select candidate enhancer-promoter interactions.

**Simulation data construction**. To simulate single-cell level RNA-seq/ATAC-seq/ HiChIP data from a mixed population with two different cell types, we downloaded the public scRNA-seq/scATAC-seq data from GM12878 and K562 and mixed them together as a single scRNA-seq/scATAC-seq dataset; we downloaded public bulk HiChIP data from GM12878 and K562 and down-sampled them as scHi-C data. In detail, we downloaded scRNA-seq and scATAC-seq, and simulated scHi-C for GM12878 and K562 separately. Then for scRNA-seq data, we computed the data matrix $E_{m \times n_1}$ where $E_{gh}$ denotes the expression level of the $g$-th gene in the $h$-th cell and $n_1 = 146$ is the total number of cells from GM12878 (73) and K562 (73). For scATAC-seq data, we computed a data matrix $O_{n \times n_2}$, where $O_{ij}$ denotes the degree of openness (i.e., read count) of the $i$-th peak in the $j$-th cell and $n_2 = 746$ is the total number of cells in GM12878 (373) and K562 (373). For scHi-C data, we computed a data matrix $C_{s \times n_3}^s$, where $C_{ij}^s$ denotes the loop counts of the $i$-th interaction in the $j$-th cell and $n_3 = 200$ is the total number of cells in GM12878 (100) and K562 (100). The numbers of cell in simulation data are shown in Table 3. We note that the scATAC-seq, the scRNA-seq data and the scHi-C data are not measured in the same cell in our setting.

**DC3 algorithm**. We first introduce some notations for our data matrices (Fig. 1b): (1) scRNA-seq matrix: $E_{m \times n1}$, where $E_{gh}$ denotes the expression level of the $g$-th gene in the $h$-th cell; (2) scATAC-seq matrix: $O_{n \times n2}$, where $O_{ij}$ denotes the degree of openness (i.e., read count) of the $i$-th enhancer in the $j$-th cell; (3) HiChIP matrix: $C_{m \times n}$, where $C_{pq}$ denotes the enhancer–promoter interactions strength (i.e., loop read counts) for the $p$-th gene's promoter and the $q$-th enhancer at the bulk level. To infer the pattern of gene expression, chromatin accessibility, and chromatin contact in each subpopulation, we formulate the following optimization problem:

$$\min_{W_1,H_1,W_2,H_2,\Lambda \geq 0} \quad \frac{1}{2}\left\|C - \alpha D \odot \left(W_1 \Lambda W_2^T\right)\right\|_F^2 + \frac{\mu_1}{2}\|E - W_1 H_1\|_F^2 + \frac{\mu_2}{2}\|O - W_2 H_2\|_F^2$$

$$s.t. \quad \sum_{k=1}^{K} h_{kj}^1 = 1; j = 1, 2, \ldots, n_1$$

$$\sum_{k=1}^{K} h_{kj}^2 = 1; j = 1, 2, \ldots, n_2$$

$$\sum_{k=1}^{K} \lambda_k = 1; \tag{1}$$

To explain this formulation, we briefly discuss each term in the objective function. (1) $\|E - W_1 H_1\|_F^2$: A soft clustering of the scRNA-seq cells can be obtained from a nonnegative matrix factorization $E = W_1 H_1$ as follows: $W_1$ has $K$ columns and $H_1$ has $K$ rows. The $i$-th column of $W_1$ gives the mean gene expression for the $i$-th cluster of cells, while the $j$-th column of $H_1$ gives the assignment weights of the $j$-th cell to the different clusters. (2) $\|O - W_2 H_2\|_F^2$: Similarly, clustering of cells in scATAC-seq data can be obtained from the factorization $O = W_2 H_2$. $W_2$ also has $K$ columns and $H_2$ has $K$ rows. The $i$-th column of $W_2$ gives the mean chromatin accessibility for the $i$-th cluster of cells, while the $j$-th column of $H_2$ gives the assignment weights of the $j$-th cell to different clusters. Note that, the $k$-th column of $W_2$ corresponds to the $k$-th column of $W_1$, indicating they are from the same cluster, namely the $k$-th cluster. (3) $\left\|C - \alpha D \odot \left(W_1 \Lambda W_2^T\right)\right\|_F^2$: We decompose each enhancer-promoter loop strength $c_{ij}$ in the bulk sample into subpopulation-specific loop strengths, i.e., $c_{ij} = \sum_k \lambda_k c_{ijk}$, where $c_{ijk}$ is the loop strength in the $k$-th subpopulation; $\lambda_k$ is proportional to the size of the subpopulation; $\Lambda$ is a $K$ by $K$ diagonal matrix $[\lambda_1, \lambda_2, \ldots, \lambda_K]$. Furthermore, based on the expectation that an enhancer-promoter loop strength is positively correlated with both the accessibilities of the enhancer and the expression values of the gene, we model $c_{ij}$ as

$$c_{ij} = \alpha d_{ij} \sum_k \lambda_k w_{ik}^1 w_{jk}^2 \tag{2}$$

Here $\alpha$ is a scaling factor; the elements $(d_{ij})$ of the matrix $D$ are indicators selecting the enhancer-promoter pair to be modeled. Only enhancer-promoter pairs with loop count larger than or equal to 1 are included into the optimization:

$$d_{ij} = \begin{cases} 1, & c_{ij} \geq 1 \\ 0, & c_{ij} < 1 \end{cases} \tag{3}$$

This leads directly to the first term in the objective function.

Finally, the objective function can be extended to handle any combination of single cell and bulk data. The general cost function is as follows:

$$\min \frac{1}{2}\alpha_1 \|C_s - W_3 H_3\|_F^2 + \frac{1}{2}\alpha_2 \sum_i \sum_{j \in RE_i} \left(C_b^{(i,j)} - \alpha_1 D_b^{(i,j)} \sum_{k=1}^{K} \lambda_k W_1^{(i,k)} W_2^{(j,k)}\right)^2$$

$$\frac{\mu_1}{2}\beta_1 \|E_s - W_1 H_1\|_F^2 + \frac{\mu_1}{2}\beta_2 \sum_i \left(E_b^{(i)} - \alpha_2 \sum_{k=1}^{K} \lambda_k \sum_{j \in RE_i} W_3^{(i,j,k)} W_2^{(j,k)}\right)^2$$

$$\frac{\mu_2}{2}\gamma_1 \|O_s - W_2 H_2\|_F^2 + \frac{\mu_2}{2}\gamma_2 \sum_j \left(O_b^{(j)} - \alpha_3 \sum_{k=1}^{K} \lambda_k \sum_{i \in TG_i} W_3^{(i,j,k)} W_1^{(i,k)}\right)^2$$

$$+ \frac{\mu_3}{2}\alpha_1 \beta_1 \gamma_1 \sum_{k=1}^{K} \lambda_k \sum_i \sum_{j \in RE_i} W_3^{(i,j,k)} W_1^{(i,k)} W_2^{(j,k)} \tag{4}$$

$$S = \left\{(i,j) \middle| C_s^{(i,j,:)} > 0 \text{ or } C_b^{(i,j)} > 0\right\}$$

$$C_b : |S| \times 1 \text{ matrix}$$

$$C_s : |S| \times n_3 \text{ matrix}$$

$$W_3 : |S| \times K \text{ matrix}$$

$$W_3^{(i,j,k)} : (i,j) \text{ represent one row in } W_3$$

$$\alpha_1 = \begin{cases} 0 & \text{indicate including HiChIP bulk input} \\ 1 & \text{indicate not including HiChIP bulk intput} \end{cases}$$

$$\alpha_2 = \begin{cases} 0 & \text{indicate including HiChIP single} - \text{cell input} \\ 1 & \text{indicate not including HiChIP single} - \text{cell intput} \end{cases}$$

$$\beta_1 = \begin{cases} 0 & \text{indicate including RNA} - \text{seq bulk input} \\ 1 & \text{indicate not including RNA} - \text{seq bulk intput} \end{cases}$$

$$\beta_2 = \begin{cases} 0 & \text{indicate including RNA} - \text{seq single} - \text{cell input} \\ 1 & \text{indicate not including RNA} - \text{seq single} - \text{cell intput} \end{cases}$$

$$\gamma_1 = \begin{cases} 0 & \text{indicate including ATAC} - \text{seq bulk input} \\ 1 & \text{indicate not including ATAC} - \text{seq bulk intput} \end{cases}$$

$$\gamma_2 = \begin{cases} 0 & \text{indicate including ATAC} - \text{seq single} - \text{cell input} \\ 1 & \text{indicate not including ATAC} - \text{seq single} - \text{cell intput} \end{cases}$$

$$s.t. \sum_{k=1}^{K} h_1^{(k,j)} = 1; j = 1, 2, \ldots, n_1$$

$$\sum_{k=1}^{K} h_2^{(k,j)} = 1; j = 1, 2, \ldots, n_2$$

$$\sum_{k=1}^{K} h_3^{(k,j)} = 1; j = 1, 2, \ldots, n_3$$

$$\sum_{k=1}^{K} \lambda_k = 1;$$

We used different $\alpha$, $\beta$, $\gamma$ to deal with different input combinations: (1) $\alpha_1 = 1$, $\alpha_2 = 0$, $\beta_1 = 1$, $\beta_2 = 0$, $\gamma_1 = 1$, $\gamma_2 = 0$, indicates scRNA-seq, scATAC-seq and scHi-C input; (2) $\alpha_1 = 0$, $\alpha_2 = 1$, $\beta_1 = 1$, $\beta_2 = 0$, $\gamma_1 = 1$, $\gamma_2 = 0$, indicates scRNA-seq, scATAC-seq and bulk HiChIP input; (3) $\alpha_1 = 0$, $\alpha_2 = 1$, $\beta_1 = 0$, $\beta_2 = 1$, $\gamma_1 = 1$, $\gamma_2 = 0$, indicates bulk RNA-seq, scATAC-seq and bulk HiChIP input; (4) $\alpha_1 = 0$, $\alpha_2 = 1$, $\beta_1 = 1$, $\beta_2 = 0$, $\gamma_1 = 0$, $\gamma_2 = 1$, indicates scRNA-seq, bulk ATAC-seq and bulk HiChIP input.

**Optimization algorithm**. We proposed a multiplicative update algorithm to solve the following non-convex optimization problem. Taking DC3 with scRNA-seq,

scATAC-seq and bulk HiChIP input as example:

$$\min_{W_1, H_1, W_2, H_2, \Lambda \geq 0} \quad \frac{1}{2}\left\|C - \alpha D \odot \left(W_1 \Lambda W_2^T\right)\right\|_F^2 + \frac{\mu_1}{2}\|E - W_1 H_1\|_F^2 + \frac{\mu_2}{2}\|O - W_2 H_2\|_F^2$$

$$s.t. \sum_{k=1}^{K} h_{kj}^1 = 1; j = 1, 2, \ldots, n_1$$

$$\sum_{k=1}^{K} h_{kj}^2 = 1; j = 1, 2, \ldots, n_2$$

$$\sum_{k=1}^{K} \lambda_k = 1;$$

$$(5)$$

Let $w_{ij}^1$ represent the element of the $i$-th row and the $j$-th column in matrix $W_1$ and $w_{ij}^2$, $h_{ij}^1$, $h_{ij}^2$ be the corresponding elements in $W_2$, $H_1$ and $H_2$. We adopted the following update scheme and stopped the iteration when the relative error was less than $10^{-4}$.

$$w_{ij}^1 \leftarrow w_{ij}^1 \frac{\left(\mu_1 E H_1^T + \alpha C W_2 \Lambda^T\right)_{ij}}{\left(\mu_1 W_1 H_1 H_1^T + \alpha^2 D \odot \left(W_1 \Lambda W_2^T\right) W_2 \Lambda^T\right)_{ij}} \quad (6)$$

$$h_{ij}^1 \leftarrow h_{ij}^1 \frac{\left(W_1^T E\right)_{ij}}{\left(W_1^T W_1 H_1\right)_{ij}} \quad (7)$$

$$w_{ij}^2 \leftarrow w_{ij}^2 \frac{\left(\mu_2 O H_2^T + C^T W_1 \Lambda\right)_{ij}}{\left(\mu_2 W_2 H_2 H_2^T + D^T \odot \left(W_2 \Lambda^T W_1^T\right) W_1 \Lambda\right)_{ij}} \quad (8)$$

$$h_{ij}^2 \leftarrow h_{ij}^2 \frac{\left(W_2^T O\right)_{ij}}{\left(W_2^T W_2 H_2\right)_{ij}} \quad (9)$$

$$h_{ij}^1 \leftarrow \frac{h_{ij}^1}{\sum_k h_{kj}^1} \quad (10)$$

$$h_{ij}^2 \leftarrow \frac{h_{ij}^2}{\sum_k h_{kj}^2} \quad (11)$$

$$\alpha \leftarrow \frac{tr(\hat{C}^T C)}{tr(\hat{C}^T \hat{C})} \hat{C} = D \odot \left(W_1 \Lambda W_2^T\right) \quad (12)$$

$$\lambda_k \leftarrow \frac{\sum_i^{n_1} h_{ik}^1 + \sum_i^{n_2} h_{ik}^2}{n_1 + n_2} \quad (13)$$

**Subpopulation-specific subnetwork connectivity (for hyper-parameter selection, see below)**. We first applied $t$-test to select top 5% subpopulation-specific genes and subpopulation-specific enhancers. Then we regarded these subpopulation-specific genes and enhancers in each subpopulation as nodes and formed $K$ subpopulation-specific subnetworks. Here we defined connectivity as the edges that fall within the given subpopulation-specific subnetwork. Suppose the subnetwork contains $n$ nodes and the strength of edge between node $i$ and $j$ is $A_{ij}$. Then the subpopulation-specific connectivity is given by the sum of $A_{ij}$ over all pairs of nodes $i,j$. Subpopulation-specific subnetwork connectivity measures the specificity of each subpopulation-specific subnetwork, including the specificity of genes, enhancers, and enhancer-gene interactions.

**Initialization and hyper-parameters selection**. We selected hyper-parameters $\mu_1$ and $\mu_2$ according to the connectivity of the subpopulation-specific subnetworks (Supplementary Table 13). We first solved the optimization problem $\min_{W_1, H_1 \geq 0}\|E - W_1 H_1\|_F^2$ and $\min_{W_2, H_2 \geq 0}\|O - W_2 H_2\|_F^2$ by the alternating least-squares (ALS) algorithm with 50 different initializations using a Monte Carlo-type approach and got the solutions for $W_{10}$, $H_{10}$, $W_{20}$, $H_{20}$, which would be used as initializations in our optimization problem. Then we calculated

$$\mu_{10} = \left\|C - \alpha D \odot \left(W_1 \Lambda W_2^T\right)\right\|_F^2 / \|E - W_1 H_1\|_F^2 \quad (14)$$

$$\mu_{20} = \left\|C - \alpha D \odot \left(W_1 \Lambda W_2^T\right)\right\|_F^2 / \|O - W_2 H_2\|_F^2 \quad (15)$$

The hyper-parameter $\mu_1$ was chosen from $\mu_{10} \times [10^0, 10^1, 10^2, 10^3, 10^4]$, and $\mu_2$ was chosen from $\mu_{20} \times [10^0, 10^1, 10^2, 10^3, 10^4]$. We used the sum the connectivity of $K$ subpopulation-specific subnetworks to select the best hyper-parameters and chose the ones which had the highest connectivity. The number of clusters $K$ can be determined by a method similar to that in Brunet et al.[36] (Supplementary Fig. 22).

**Subpopulation profiles**. For single-cell input, we calculated the mean profiles for those cells with the same cluster labels. For example, the subpopulation gene expression profiles are given by the columns of the matrix $P_1$, The $k$-th column of $P_1$ is computed by averaging the single cell expression profile of the cells in cluster $K$. For example, $P_1^{(\cdot k)} = \sum_{j \in S_k} E_{\cdot j}/|S_k|$, $S_k:\{j|j - th \text{ cell belongs to cluster } K\}$. The cluster mean based subpopulation profile $P_1$ is similar to the subpopulation profile $W_1$ from matrix factorization. However, the results on simulation data show that

the cluster mean based subpopulation profile have better performance in HiChIP deconvolution than the matrix factorization-based subpopulation profile. The computation of subpopulation profile $P_2$ from scATAC-seq and subpopulation profile $P_3$ from scHi-C are similar.

For bulk data type, we can use a simple plug-in expression to obtain its subpopulation profiles from the subpopulation profiles already obtained from single-cell data. For example, the subpopulation HiChIP profiles $P_3$ can be obtained from single-cell-averaged- profiles $P_1$ and $P_2$ by using the expression $p_{ijk}^3 = \alpha \hat{d}_{ij} \lambda_k p_{ik}^1 p_{jk}^2$.

**Poisson model-based estimate for subpopulation HiChIP profiles**. We also developed a statistical model for the deconvolution of bulk HiChIP profile into subpopulation HiChIP profiles.

In this model the observed loop count $C_{ij}$, is a sum of latent loop counts, i.e.,

$$C_{ij} = \sum_k C_{ijk} \quad (16)$$

where $C_{ijk}$ indicates the loop counts between the $i$-th promoter and the $j$-th enhancer in the $k$-th cluster.

We assume that $C_{ijk}$ is generated from a Poisson model

$$C_{ijk} \sim Poisson\left(\lambda \sqrt{O_{ik} O_{jk}}\right) \quad (17)$$

$O_{ik}$ indicates the openness of the $i$-th promoter in the $k$-th cluster, $O_{jk}$ indicates the openness of the $j$-th enhancer in the $k$-th cluster, and these openness values are assumed to be known constants. Let

$$\boldsymbol{M_{ij}} = \left(C_{ij1}, \ldots, C_{ijK}\right) \quad (18)$$

$$\boldsymbol{\rho_{ij}} = \left(\rho_{ij1}, \ldots, \rho_{ijK}\right) \quad (19)$$

$$\rho_{ijk} = \frac{\sqrt{O_{ik} O_{jk}}}{\sum_k \sqrt{O_{ik} O_{jk}}} \quad (20)$$

Then

$$\boldsymbol{M_{ij}}|C_{ij} \sim Multinomial\left(C_{ij}, \boldsymbol{\rho_{ij}}\right) \quad (21)$$

Let $E_{ik}$ be the gene expression of the $i$-th gene in the $k$-th cluster. We assume this gene expression follows a Poisson distribution with a rate proportional to the sum of cluster-specific HiChIP interactions involving the $i$-th promoter.

$$E_{ik} \sim Poisson\left(\beta \sum_q C_{iqk}\right) \quad (22)$$

Given the two sets of observations $\{C_{ij}\}$ and $\{E_{jk}\}$, our task is to infer the set of latent variables $\{C_{ijk}\}$. We do this by iteratively computing the MAP (maximum a posteriori) estimate of the latent multinomial variable $\boldsymbol{M_{ij}}$ conditional on the set of all other latent variables $\{\boldsymbol{M_{rs}}:(r, s) \neq (i, j)\}$. Specifically, given the current values of $\{C_{rsk}:(r, s) \neq (i, j)\}$ we compute the value of $\{C_{ij1}, \ldots, C_{ijK}\}$ that maximizes the following conditional posterior probability:

$$\binom{C_{ij}}{\boldsymbol{M_{ij}}} \prod_{k=1}^{K} \rho_{ijk}^{C_{ijk}} e^{-\beta\left(\sum_{q \neq j} C_{iqk} + C_{ijk}\right)} \left[\beta\left(\sum_{q \neq j} C_{iqk} + C_{ijk}\right)\right]^{E_{ik}} \quad (23)$$

**Dropout simulation**. Dropout usually refers to the phenomenon that an expressed RA molecule might not be captured in a single cell. To test whether our algorithm could still function well in the presence of dropout events, we used a Bernoulli distribution to decide which "sites" (genes or enhancers) should be dropped in scRNA-seq and scATAC-seq data. Zero values were introduced into the simulated data for each gene/enhancer based on a Bernoulli distribution defined by the dropout rate. In our experiments, we chose the dropout rate from [0, 0.5, 0.8, 0.9].

**Down sampling scRNA-seq and scATAC-seq in RA day 4**. To simulate dropout based on scRNA-seq dataset from Drop-seq platform, we first down sampled each gene's read count or each enhancer's read count $P_{ij}$ as $\hat{P}_{ij} = P_{ij}/100$, where $\hat{P}_{ij} \sim Poi(P_{ij}/100)$, and the dropout effect was modeled as $D_i \sim Ber\left(\frac{1}{1 + P_{ij}/100^{-0.1}}\right)$ ($P$ indicates the openness matrix $O$ or expression matrix $E$).

**Surface markers selection**. To sort the subpopulation 2 cells, we selected subpopulation 2 specific surface markers from gene expression data. We required that the selected surface markers satisfy the following conditions: (i) Differentially expressed between subpopulation 2 and the other subpopulations. (ii) Highly expressed in subpopulation 2, and (iii) expression level in subpopulations 1 and 3 are less than 2. In practice, we compared the distribution of surface markers' expression in subpopulation 2 versus the other subpopulations by $t$-test. We selected the top 20 markers and further require that TPM expression in subpopulation 2 be greater than 10 and higher than in subpopulations 1 and 3.

**Performance evaluation**. There are two tasks for our algorithm: (1) deconvolution of subpopulation-specific HiChIP; (2) coupled clustering of scRNA-seq and scATAC-seq. For deconvolution, we ran our algorithm 50 times and evaluated the results in terms of mean Pearson Correlation Coefficient (PCC) of true subpopulation-specific HiChIP values and the predicted values. In detail, if there are $n$ interactions in the bulk HiChIP data, both the true subpopulation-specific HiChIP and the predicted subpopulation-specific HiChIP are represented using $n$-dimensional vectors. Then the deconvolution performance is evaluated by calculating the PCC score between the true vector and predicted vector. For coupled clustering, we evaluated the performance in terms of error rate of true subpopulation labels and the predicted cluster assignments. We ran our algorithm and NMF 50 times from different initial values to calculate the mean error rate and compared our algorithm with NMF.

**Subpopulation-specific genes**. We defined subpopulation-specific genes according to its $p$-values from scRNA-seq and HiChIP data. For scRNA-seq, we applied a one-tailed $t$-test to define the subpopulation-specific genes and obtained the scRNA-seq $p$-values; For HiChIP, we first applied a one-tailed binomial test to define subpopulation-specific interactions. To eliminate the bias of various loop counts of interactions, we normalized the total loop counts to N (e.g., $N = 10$) for each interaction and get the modified loop counts in the $k$-th subpopulation as $n_k$. The expected proportion of the interactions in each subpopulation $p_k$ is regarded as $1/K$ based on the assumption that each interaction is uniformly distributed in each subpopulation. Then, we calculated $p$-value for the interaction in the $k$-th subpopulation using the binomial test in R binom.test ($n_k$, $N$, $p_k$, alternative = "greater"). For genes with more than one interaction, we chose the most significant $p$-value as the HiChIP $p$-value of the gene. We further combined these two $p$-values for each gene using Fisher's methods and select the top 1000 subpopulation-specific genes with the smallest combined $p$-values in each subpopulation.

**Motif enrichment scores**. We merged the scATAC-seq cells in each subpopulation to get a merged sample. On this sample, we used MACS2 to call the open peaks. We performed motif enrichment analysis on those open peaks by Homer. The motif enrichment score was defined by geometric mean of $-\log10(p\text{-value})$ and fold change.

$$\text{Motif enrichment score} = \sqrt{-\log_{10}(\text{pvalue}) \times \text{FoldChange}} \qquad (24)$$

**GO terms selection**. For each subpopulation, we ranked GO terms using motif enrichment scores and kept the significant GO terms with scores larger than 2. Then we removed the GO terms which were significant in all three subpopulations. For example, GO terms cell projection morphogenesis, regulation of cellular component movement, regulation of localization, regulation of biological quality and etc. were significant in all three subpopulations, and we removed these GO terms for subsequent subpopulation-specific analysis.

**Dense subnetwork detection**. Given a directed weighted graph $(G, W)$, we obtained the dense subnetwork by solving the following optimization problem:

$$\max \sum_i \sum_j W_{ij} x_i y_j$$
$$\text{s.t.} x_1^\beta + x_2^\beta + \dots + x_n^\beta = 1$$
$$y_1^\beta + y_2^\beta + \dots + y_n^\beta = 1 \qquad (25)$$
$$x_i \geq 0; i = 1, 2, \dots n$$
$$y_i \geq 0; i = 1, 2, \dots n$$

where nonzero value of $x_i$ indicates that the $i$-th node in graph $G$ is included in the subnetwork as a regulator, nonzero value of $y_i$ indicates that the node is included in the module as a target. We chose $\beta = 1$, which leads to L1-type constraint that promotes sparse solutions. Then, the dense subnetwork was given by the set of nodes with nonzero $x_i$ or $y_i$. We can further partition the dense subnetwork into the core node $C$, the upstream node $U$ and the downstream node $D$, defined respectively as

$$U = \{i | x_i > 0, y_i = 0\} \qquad (26)$$

$$C = \{i | x_i > 0, y_i > 0\} \qquad (27)$$

$$D = \{i | x_i = 0, y_i > 0\} \qquad (28)$$

To test whether the extracted dense subnetwork is statistically significant or not, we generated a null distribution by permuting the network. In the permutation, to maintain the same in-degree and out-degree of each nodes of the network, we used the switching permutation operation (selected two edge every time and switch source and target), and switched 1000 times to generate the random network. We generated 1000 random networks and extracted the dense subnetwork on random networks. We calculated a $p$-value by comparing the optimal value from our network with that from random networks.

**Cell culture**. Mouse ES cell lines R1 were obtained from American Type Culture Collection (ATCC, Cat. no. SCRC-1036). The mESCs were first expanded on an MEF feeder layer previously irradiated. Then, subculturing was carried out on 0.1% bovine gelatin-coated tissue culture plates. Cells were propagated in mESC medium consisting of Knockout DMEM supplemented with 15% knockout serum replacement, 100 μM nonessential amino acids, 0.5 mM beta-mercaptoethanol, 2 mM GlutaMax, and 100 U/mL Penicillin-Streptomycin with the addition of 1000 U/mL of LIF (ESGRO, Millipore).

**Cell differentiation**. mESCs were differentiated using the hanging drop method[37]. Trypsinized cells were suspended in differentiation medium (mESC medium without LIF) to a concentration of 50,000 cells/ml. 20 μl drops (~1000 cells) were then placed on the lid of a bacterial plate and the lid was upside down. After 48 h incubation, Embryoid bodies (EBs) formed at the bottom of the drops were collected and placed in the well of a 6-well ultra-low attachment plate with fresh differentiation medium containing 0.5 μM retinoic acid (RA) for up to 4 days, with the medium being changed daily.

**HiChIP**. We followed the HiChIP protocol published by Mumbach et al.[37], using antibody to H3K27ac (Abcam, ab4729) with the following modifications. The EBs were first treated with StemPro Accutase Cell Dissociation Reagent (Thermo Fisher) at 37 °C for 10–15 min with pipetting. Approximately one million cells were crosslinked with freshly prepared 1% formaldehyde. The pellet was then resuspended in 500 μl of ice-cold Hi-C Lysis buffer. After digestion with 25 U (5 μl of 5U/μl) MboI restriction enzyme and ligation, the nuclear pellet was brought up to 880 μl of Nuclear Lysis Buffer. Samples were sheared using a Covaris E220 using the following parameters: fill level = 10, duty cycle = 5, PIP = 140, cycles/burst = 200, time = 2 min and then clarified by centrifugation for 15 min at 16,100 × g at 4 °C. The samples were precleared with 6 μl Dynabeads Protein A (Thermo Fisher) at 4 °C for 1 h. We then added 2.5 μg of antibody to H3K27ac, and captured the chromatin-antibody complex with 6 μl of Dynabeads Protein A. Approximately 2–4 ng of ChIP DNA was obtained following Qubit quantification. The amount of Tn5 used and number of PCR cycles performed were based on the post-ChIP Qubit amounts, as described in the HiChIP protocol[37]. The library was sequenced on Illumina NextSeq 500 with 75 bp paired-end reads. Total 13 million cells were used in HiChIP experiment.

**Fluorescence-activated cell sorting (FACS)**. The EBs treated with RA for 4 days were trypsinized with 1 ml StemPro Accutase Cell Dissociation Reagent (Thermo Fisher) at 37 °C for 10–15 min with pipetting. Once EBs got dissociated, 4 ml of Flow Cytometry Staining Buffer (Invitrogen, Cat. no. 00-4222) was add to the cell sample. The single cells were obtained by filtering twice with 40 μm cell strainer. After centrifuge at 500 × g for 4 min, the supernatant was removed and cells were resuspended in 500–700 μl of Flow Cytometry Staining Buffer to obtain the final concentration of $4 \times 10^7$ cells/ml. 100 μl cells were used as unstained negative control cells for FACS analysis. The remaining cells were distributed at 100 μl per tube (~$4 \times 10^6$ cells) into Falcon® 12 × 75 mm round-bottom polystyrene test tube (Thermo Scientific), 100 μl per tube. To block non-specific Fc-mediated interactions, all tubes were first pre-incubated with 0.5 μg of Anti-Mouse CD16/32 antibody (1:40 dilution, Invitrogen, Cat. no. 14-0161) for 15 min at 4 °C. Then 0.125 μg PE-Cy7-labeled EpCAM (1:160 dilution, Invitrogen, Cat. no. 25-5791-80) and 0.1 μg PE-labeled CD38 (1:200 dilution, Invitrogen, Cat. no. 12-0381-82) were added to the tubes. After incubation for at least 30 min on ice, the cells were washed with 2 ml Flow Cytometry Staining Buffer + 1 mM EDTA to prevent cell adhesion. The cells were spin down at 500 × g for 5 min at room temperature and the wash step was repeated twice. The final cells were resuspended in 200 μl of Flow Cytometry Staining Buffer + 1 mM EDTA for FACS analysis. As compensation controls, 1 drop of UltraComp eBeads (Invitrogen, Cat. no. 01-2222) was added to three empty 12 × 75 mm round bottom test tubes, followed by adding 0.125 μg PE-Cy7-labeled EpCAM (labeled as PE-Cy7 only compensation beads), 0.1 μg PE-labeled CD38 (labeled as PE only compensation beads), or no antibody (labeled as no stain compensation beads). After mixing well by flicking, the tubes were incubated on ice for 20 min, followed by washing with 2 ml of Flow Cytometry Staining Buffer twice. After removing supernatant, 200 μl from each tube was used as compensation controls for FACS analysis. Five independent experiments were performed for FACS analysis, each time we obtained approximately 700,000 EpCAM and CD38 double positive cells. Those cells were collected in 15 ml conical tube, then the cells were spin down and crosslinked with freshly prepared 1% formaldehyde based on the HiChIP protocol[37]. After crosslinking, the cells were ready for the following HiChIP analysis.

**Software availability**. DC3 is implemented in Python 2.7 and freely available at https://github.com/SUwonglab/DC3.

**Reporting summary**. Further information on research design is available in the Nature Research Reporting Summary linked to this article.

## Data availability

The HiChIP data that support the findings of this study have been deposited in Gene Expression Omnibus (GEO) with the accession code GSE127807. The single-cell data that support the finding of this study are available in GEO with the accession code GSE115968 and GSE107651. All other relevant data supporting the key findings of this study are available within the article and its Supplementary Information files or from the corresponding authors upon reasonable request. The source data underlying Fig. 2a, b and 3 are provided as a Source Data file. A reporting summary for this Article is available as a Supplementary Information file.

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

## Acknowledgements

W.H.W. and Z.D. were supported by NIH grants P50HG007735, R01HG010359, and R01GM109836. X.C. was partially supported by VA Palo Alto Health Care System. Y.W. was supported by the Strategic Priority Research Program of the Chinese Academy of Sciences (XDB13000000). W.W.Z. and R.J. were supported by National Key Research and Development Program of China No. 2018YFC0910404, the National Natural Science Foundation of China Nos. 61873141, 61721003, 61573207, and the Tsinghua-Fuzhou Institute for Data Technology. R.J. is also supported by a RONG professorship at the Institute for Data Science of Tsinghua University. Cell sorting/flow cytometry analysis for this project was done on instruments in the Stanford Shared FACS Facility.

## Author contributions

W.H.W. and R.J. conceived the project. W.W.Z. and Z.D. designed the analytical approach and performed numerical experiments and data analysis. X.C. performed all biological experiments. W.W.Z. wrote the software. Y.W., R.J., and W.H.W. supervised the research. All authors wrote, revised, and contributed to the final paper.

## Competing interests

The authors declare no competing interests.
