## [Peer Review File · Nature Communications]

Reviewers' comments:

Reviewer #1 (Remarks to the Author):

This manuscript describes a method to jointly analyze scRNA, scATAC and bulk HiChIP data. This type of joint analysis of heterogeneous single-cell data is becoming increasingly common, so this manuscript addresses an important problem. On the other hand, the application here is quite specific: the authors demonstrate how their approach works for a particular combination of three data types. As alluded to in the discussion, it may be more sensible to use scHi-C rather than bulk Hi-C in this setting. The method is a modification of this group's previous technique for the analysis of scRNA and scATAC data. Overall, the results are fairly convincing, with some important caveats outlined below. As a whole, the manuscript would be more compelling if it provided a general framework for working with various combinations of bulk and single-cell data types.

The first section of Results reads like a Methods section rather than a Results section. Rather than simply reporting on what the method consists of, the text should clearly provide the motivation for this particular formulation and make explicit the assumptions that underlie the model. The model underlying the first term in the objective seems particularly arbitrary. Empirical evidence to support this part of the model should be provided. Furthermore, a major drawback of the method is that it assumes that one can identify all of the enhancers a priori. The section should also explicitly enumerate all the hyperparameters of the model. The Methods section outlines how these hyperparameters are selected, but this should be described briefly in the main document.

line 23 and line 50: It is not actually that difficult to measure Hi-C contacts in single cells via combinatorial indexing.

line 85: The dimensionality of W_2 is not specified. Does it have K columns, and is there a constraint that these correspond to the same clusters as in W_1 ?

line 95: Discarding all pairs with loop count < 2 seems fairly arbitrary, especially as it ignores the genomic distance effect.

The section on simulations is reasonable. However, more interesting negative controls should be created, keeping two of the three data sets the same and permuting the third data set.

line 125-126: A Pearson correlation is computed between "loop profiles." This terminology is not adequately precise. Exactly how the correlation is computed must be spelled out.

The results in Supplementary Figures 5-6 should be alluded to in Figure 2B as solid lines (median) and flanking error bars (interquartile range).

I did not find the section about how DC3 helps interpret subpopulations very convincing. For example, the results in Figure A and the corresponding text (lines 169-173) are problematic because I don't believe that you can compare these Pearson correlations to draw conclusions about which type of data

is more cell-type specific. Pearson correlations for HiChIP data are naturally high because of the genomic distance effect. More fundamentally, these are correlations over vectors of different lengths and computed across fundamentally different types of data.

The caption to Figure 3B (erroneously labeled E in the figure) fails to explain what the numbers in the table are. Furthermore, the table and the text do not specify how these GO terms were selected; i.e., are these all the terms deemed significant at a specified confidence level? It seems surprising that there are precisely three terms in each case.

line 152: Why was subpopulation 2 selected for further study?

Figure 2C should include a color scale.

Figure 2D needs to be explained better. It is not clear exactly what values went into the PCA.

The three-step protocol outlined in the final section of Results seems reasonable but is not particularly well validated. The only validation provided is the qualitative assessment of the TFs enriched in subpopulations 1 and 2 (lines 236-244).

line 190: "as shown by the enrichment p-values" Where can I find these p-values? Maybe those are the numbers in the table, though they presumably must have been negative log transformed.

line 260-261: We are told that the hyperparameters of DC3 can be tuned automatically, but the manuscript does not provide any evidence to show that this tuning is effective.

line 264: It is debatable whether the purported improvements demonstrated here can accurately be described as "dramatic."

Reviewer #2 (Remarks to the Author):

The authors present a computational method to integrate scRNA-seq, scATAC-seq and bulk HiChIP data. The method simultaneously deconvolves and clusters the data.

This is an interesting method and seems useful given that most experimental technologies can only be done in bulk while only some technologies in single cells. Combining bulk with single cell data thus seems a promising direction.

Comments:

It is not that common to have scRNA-seq, scATAC-seq as well as HiChIP seq data for the same system. It would be useful if this method, or some version of it, would work in the case that for example only scRNA-seq data and various bulk data are available, which would be a situation that is more common. So, the method would have more impact if it were to be presented as a general deconvolution method; one that not only works on the presented data types.

I would like to see more robustness and validation analyses. These do not have to be experimental validation, however can be using simulated data. For robustness, it would be good to show that the

results are the same when downsampling cells.

Is the result robust to the number of factors (in nmf) chosen?

Typos:

Line 150: "We now want [to] assess"

Line 152: "We focussed on one [of] the subpopulation[s]"

Line 159: "we performed [a] HiChIP"

Point-by-point responses to Reviewer 1

Comment: This manuscript describes a method to jointly analyze scRNA, scATAC and bulk HiChIP data. This type of joint analysis of heterogeneous single-cell data is becoming increasingly common, so this manuscript addresses an important problem..... Overall, the results are fairly convincing, with some important caveats outlined below.

Response: Thank you for this positive remark. We are also very grateful for your detailed comments and helpful suggestions.

Comment: As a whole, the manuscript would be more compelling if it provided a general framework for working with various combinations of bulk and single-cell data types.

Response: Following your suggestion, we have extended our method to handle various combination of single cell and bulk data. In the original version, DC3 can only handle one setting of input data combinations, namely (scRNA-seq, scATAC-seq, bulk HiChIP). In this revision, we developed a more general formulation of the optimization problem where, for each type of measurement, our cost function will include a NMF term and/or a coupling term depending on whether single cell or bulk data (or both) are available. In this way, the optimization can be used to handle almost any combination of input data.

Comment: The first section of Results reads like a Methods section rather than a Results section. Rather than simply reporting on what the method consists of, the text should clearly provide the motivation for this particular formulation and make explicit the assumptions that underlie the model.

Response: Following your suggestion, we have moved the exact mathematical formulation of the DC3 algorithm to the Methods section. Instead of presenting the technical details, in the Results section we begin by discussing the motivation of our approach and the meaning of the terms in our general cost function. We then assessed the method's performance by simulation. Using simulated mixtures of K562 and GM12878 cells, we performed a systematic study under the following four input settings: 1) scRNA-seq, scATAC-seq, scHiChIP; 2) scRNA-seq, scATAC-seq, bulk HiChIP; 3) scRNA-seq, bulk ATAC-seq, bulk HiChIP; 4) bulk RNA-seq, scATAC-seq, bulk HiChIP. The results showed that with high sequencing depth, all four settings could yield good estimates of subpopulation-specific profiles for expression, accessibility, and enhancer-promoter contact. With low sequencing depth (i.e., high dropout rate), the estimates from settings 1 & 2 remained satisfactory while those from settings 3 & 4 were quite poor. After introducing and discussing the general formulation, in the remainder of the paper we focused only on setting 2 to further analyze DC3's performance relative to alternative methods, and to demonstrate the methodology in a real data application.

Comment: The model underlying the first term in the objective seems particularly arbitrary. Empirical evidence to support this part of the model should be provided.

Response: Thank you for raising this excellent point, which prompted us to think harder about the purpose of the coupling term in the cost function. Our current view is that this coupling term does not need to have a rigorous interpretation based on a generative statistical model. Rather, as long as this term (first term of Figure 1B) is able to enforce certain statistical relationships among the three types of data, then it may be used to couple the two NMF- clustering computations (of the scRNA-seq cells and the scATAC-seq cells). Since in homogenous populations (cell lines) we have observed positive correlations (Supplementary Figures 1 & 2) i) between HiChIP loop count and gene expression, and ii) between HiChIP and accessibility, we designed the coupling term to induce a bilinear dependency of HiChIP signal on expression and accessibility. Our simulation results confirmed that this coupling led to improved clustering results in both sc samples. From this perspective, the optimization allowed us to obtain better clustering results, but it may or may not provide the best subpopulation profiles.

Investigation on this question led us to make a major change in the way we estimate the subpopulation profiles. Originally, after solving the optimization, we read off the subpopulation-specific profiles for expression and accessibility directly from the columns of the W_1 and W_2 matrices, and these profiles were then used to compute the subpopulation-specific HiChIP profile based on a simple expression. In this revision, instead of using the W matrices, we obtained the subpopulation profile for expression (or accessibility) by averaging the expression (or accessibility) profiles of the single cells that has been assigned (by the H_1 or H_2 matrices) to that subpopulation. These new estimates were then used to compute the subpopulation HiChIP profile. In simulated mixtures, this new method is seen to offer substantial improvement over the original method. In addition to the above simple method, we also investigated an alternative method, based on a Poisson model for the (unobserved) subpopulation loop counts, to infer the subpopulation HiChIP profile from its conditional posterior distribution. The details of this model is given in Methods. While the Poisson model-based method is more rigorous statistically and also offers slightly better performance in simulated mixtures, the improvement is always quite small (Supplementary Figures 3 and 4). Because the simple method is much faster computationally, we decided to use it as the default method to estimate subpopulation HiChIP profiles.

Comment: Furthermore, a major drawback of the method is that it assumes that one can identify all of the enhancers a priori.

Response: We assume that the scATAC-seq data, when pooled together, can be used to call peaks and identify candidate enhancers in all subpopulations. Then, DC3 will aim to deconvolute the bulk HiChIP interactions restricted to these candidate enhancers. Therefore, without need of a pre-defined enhancer set, DC3 can define candidate enhancers using the scATAC-seq and HiChIP data directly. Of course, if the pooled scATAC-seq data is not deep enough, there is the possibility of missing some enhancers in the smaller subpopulations. In such cases, it may be worthwhile to supplement enhancers by adding some known enhancers from relevant cell lines.

Comment: The section should also explicitly enumerate all the hyperparameters of the model. The Methods section outlines how these hyperparameters are selected, but this should be described briefly in the main document.

Response: Following your suggestion, we have underlined the definition of candidate enhancer sets in “Results: The DC3 algorithm”. In addition, we also added description of the hyperparameters which can be tuned automatically in the same section.

Comment: line 23 and line 50: It is not actually that difficult to measure Hi-C contacts in single cells via combinatorial indexing.

Response: We have revised the description of single cell Hi-C and added the following statements:

“Furthermore, while currently scHi-C can directly measure the chromatin contact (i.e. loop) between active regulatory elements to their target genes at the single cell level, bulk HiChIP can measure such contacts in a higher resolution.”

“In bulk sample it is easy to measure 3D contacts between active enhancers and gene promoters by H3K27ac HiChIP experiments, while combinatorial indexing has also been used for 3D contact measurement in single cells.”

Comment: line 85: The dimensionality of W_2 is not specified. Does it have K columns, and is there a constraint that these correspond to the same clusters as in W_1 ?

Response: Both W_1 and W_2 have K columns, and the k -th column of W_2 corresponds to the k -th column of W_1 , indicating they are from the same cluster, namely the k -th cluster. We have added sentences to clarify this point in the Methods section.

Comment: line 95: Discarding all pairs with loop count < 2 seems fairly arbitrary, especially as it ignores the genomic distance effect.

Response: Thank you for your advice. Plotting contact probability as a function of distance illustrates that contact probability scales as a power law with genomic distance whose slope is approximately -1^1 , so the contact probabilities of long-distance pairs are much smaller than the short-distance ones. Following your suggestion, we now do not discard any pairs in this version since long-distance pairs with small loop counts may also be important. We changed the elements (d_{ij}) of the matrix D as following:

$$d_{ij} = \begin{cases} 1, & c_{ij} \geq 1 \\ 0, & c_{ij} < 1 \end{cases}$$

The change does not have noticeable impact on the results.

Comment: The section on simulations is reasonable. However, more interesting negative controls should be created, keeping two of the three data sets the same and permuting the third data set.

Response: Since we did not use any label information from scRNA-seq, scATAC-seq and HiChIP in our objective function, permuting the label did not affect the results of DC3. Instead,

we provided three negative controls, by keeping two of the three data sets the same and permutating third data randomly (Supplementary Tables 5-7). From these negative controls, each data type is important to both joint clustering and deconvolution. We have added these negative controls results in “Results: Performance evaluation of deconvolution on in silico mixture of cells” and “Results: Performance evaluation of deconvolution on experimental mixtures” in this version.

Comment: *line 125-126: A Pearson correlation is computed between loop profiles." This terminology is not adequately precise. Exactly how the correlation is computed must be spelled out.*

Response: We have added detailed description of how the correlation is computed in “Methods: Performance evaluation” as following:

“In detail, if there are n interactions in the bulk HiChIP data, both the true subpopulation-specific HiChIP and the predicted subpopulation-specific HiChIP are represented using n -dimensional vectors. Then the deconvolution performance is evaluated by calculating the PCC score between the true vector and predicted vector.”

Comment: *The results in Supplementary Figures 5-6 should be alluded to in Figure 2B as solid lines (median) and flanking error bars (interquartile range).*

Response: We have updated Figure 2A (original Figure 2B) with the random deconvolution results according to your suggestion.

Comment: *I did not find the section about how DC3 helps interpret subpopulations very convincing. For example, the results in Figure 3A and the corresponding text (lines 169-173) are problematic because I don't believe that you can compare these Pearson correlations to draw conclusions about which type of data is more cell-type specific. Pearson correlations for HiChIP data are naturally high because of the genomic distance effect. More fundamentally, these are correlations over vectors of different lengths and computed across fundamentally different types of data.*

Response: We agreed with your opinion and have removed the Figure 3A and the comparison of different data types using the PCC scores in this version.

Comment: *The caption to Figure 3B (erroneously labeled E in the figure) fails to explain what the numbers in the table are. Furthermore, the table and the text do not specify how these GO terms were selected; i.e., are these all the terms deemed significant at a specified confidence level? It seems surprising that there are precisely three terms in each case.*

Response: We apologize for the error in the label. In this version, we have updated the legend of Figure 3B to “B) Subpopulation-specific GO terms enrichment results. The enrichment p -values

are transformed to $-\log_{10}(pvalue)$ and shown in the table. Original: scRNA-seq measured in SMART-seq with median ~1 million reads per cell; Down-sampling: simulated scRNA-seq measured in Drop-seq with median UMI ~5,000.”

To answer your question on how the GO terms are selected: For each subpopulation, we ranked GO terms using motif enrichment scores, which were defined by geometric mean of $-\log_{10}(pvalue)$ and fold change, and kept the significant GO terms with scores larger than 2.

$$\text{motif enrichment score} = \sqrt{-\log_{10}(p - value) \times FoldChange}$$

Then we removed the GO terms which were significant in all three subpopulations. For example, GO terms “cell projection morphogenesis”, “regulation of cellular component movement”, “regulation of localization” “regulation of biological quality” and etc. were significant in all three subpopulations, and we removed these GO terms for further subpopulation-specific analysis.

Comment: line 152: Why was subpopulation 2 selected for further study?

Response: Consistent with our previous results², subpopulation 1 and subpopulation 3 are two related subpopulations. This can be seen in the clustering stability in RA day 4 single cell data for $K = 2$ to 5 (See Supplementary Figure 23). When $K = 2$, subpopulation 1 and subpopulation 3 are merged into one subpopulation, which indicates these two subpopulations tend to share more similar patterns. In order to isolate pure cell population for downstream experimental validation, we decided to select the most distinct subpopulation, namely subpopulation 2. To clarify this issue, we have added the reason for selecting subpopulation 2 in “Result: Performance evaluation of deconvolution on experimental mixtures”.

Comment: Figure 2C should include a color scale.

Response: Thank you for your advice. We have added the color scale Figure 2C.

Comment: Figure 2D needs to be explained better. It is not clear exactly what values went into the PCA.

Response: We have added detailed description of the legend of Figure 2D as following:

“D) Performance of HiChIP deconvolution in RA-day 4 real data. The HiChIP profile measured from double positive cells is much closer to that inferred for subpopulation 2 than to the HiChIP profiles inferred for the other subpopulation or measured from the bulk sample. All HiChIP profiles are represented using n -dimensional vectors with each dimension indicating corresponding loop counts.”

Comment: The three-step protocol outlined in the final section of Results seems reasonable but is not particularly well validated. The only validation provided is the qualitative assessment of the TFs enriched in subpopulations 1 and 2 (lines 236-244).

Response: We regard the three-step protocol as an exploratory tool to help us extract and

visualize useful information from the numerous (TF-RE-TG) triplet regulatory relations provided by the cluster specific profiles. In our 3-step protocol, the first step selects important TFs as those with significant motif enrichment in open regions. The TF-RE-TG relations involving these TFs are still very large in number, so in the second step we extract TF-TG pairs by integrating over REs in the relevant triplets. This gives rise to a directed (TF-TG) network which is smaller but still too large for visualization. Thus in the third step, using mathematical programming we look for an even smaller subnetwork of genes with a highly elevated number of regulatory relations among them. Intuitively, such a “densest subnetwork” is likely to reveal important regulatory modules composing of important TFs regulating each other and their downstream TGs. Our qualitative assessment of the resulting TFs in each subpopulation in RA day 4 seems to support the importance of dense networks.

Comment: line 190: 'As shown by the enrichment p-values' Where can I find these p-values? Maybe those are the numbers in the table, though they presumably must have been negative log transformed.

Response: Thank you for this point. We have changed the description of enrichment p-values to $-\log_{10}(p - \text{value})$, which is now consistent with Table in Figure 3. For example, the number 37.83 in the table means that the p-value equals to $10^{-37.83} = 1.48 \times 10^{-38}$.

Comment: line 260-261: We are told that the hyperparameters of DC3 can be tuned automatically, but the manuscript does not provide any evidence to show that this tuning is effective.

Response: Thank you this question. To address it, we have added the detailed hyperparameters selection results in simulation study in “Methods: Initialization and parameters selection” and Supplementary Figure 22. Briefly, our approach is to use the sum connectivity of the K subpopulation-specific subnetworks to select the best parameters and chose the ones which had the highest connectivity.

Comment: line 264: It is debatable whether the purported improvements demonstrated here can accurately be described as dramatic."

Response: We agree and have removed the word “dramatically” in this version.

Reference

1. van Berkum, N.L. *et al.* Hi-C: a method to study the three-dimensional architecture of genomes. *J Vis Exp* (2010).
2. Duren, Z. *et al.* Integrative analysis of single-cell genomics data by coupled nonnegative matrix factorizations. *Proc Natl Acad Sci U S A* **115**, 7723-7728 (2018).

Point-by-point responses to Reviewer 2

Comment: *The authors present a computational method to integrate scRNA-seq, scATAC-seq and bulk HiChIP data. The method simultaneously deconvolves and clusters the data. This is an interesting method and seems useful given that most experimental technologies can only be done in bulk while only some technologies in single cells. Combining bulk with single cell data thus seems a promising direction.*

Response: Thank you for the positive remark. We are grateful for your detailed comments and helpful suggestions.

Comment: *It is not that common to have scRNA-seq, scATAC-seq as well as HiChIP seq data for the same system. It would be useful if this method, or some version of it, would work in the case that for example only scRNA-seq data and various bulk data are available, which would be a situation that is more common. So, the method would have more impact if it were to be presented as a general deconvolution method; one that not only works on the presented data types.*

Response: Following your suggestion, we have extended our method to handle various combination of single cell and bulk data. In the original version, DC3 can only handle one setting of input data combinations, namely (scRNA-seq, scATAC-seq, bulk HiChIP). In this revision, we developed a more general formulation of the optimization problem where, for each type of measurement, our cost function will include a NMF term and/or a coupling term depending on whether single cell or bulk data (or both) are available. In this way, the optimization can be used to handle almost any combination of input data. The exact mathematical formulation of the extended DC3 algorithm is given in the Methods section. Instead of presenting the technical details, in the Results section we begin by discussing the motivation of our approach and the meaning of the terms in our general cost function. We then assessed the method's performance by simulation. Using simulated mixtures of K562 and GM12878 cells, we performed a systematic study under the following four input settings: 1) scRNA-seq, scATAC-seq, scHiChIP; 2) scRNA-seq, scATAC-seq, bulk Hi-ChIP; 3) scRNA-seq, bulk ATAC-seq, bulk HiChIP; 4) bulk RNA-seq, scATAC-seq, bulk HiChIP. The results showed that with high sequencing depth, all four settings could yield good estimates of subpopulation-specific profiles for expression, accessibility, and enhancer-promoter contact. With low sequencing depth (i.e., high dropout rate), the estimates from settings 1 & 2 remained satisfactory while those from settings 3 & 4 were quite poor. After introducing and discussing the general formulation, in the remainder of the paper we focused only on setting 2 to further analyze DC3's performance relative to alternative methods, and to demonstrate the methodology in a real data application.

Comment: *I would like to see more robustness and validation analyses. These do not have to be experimental validation, however can be using simulated data. For robustness, it would be good to show that the results are the same when downsampling cells.*

Response: Thank you for your suggestion. Following your advice, we further conducted a series of experiments for downsampling cells, where the numbers of cell for both scRNA-seq and scATAC-seq data were downsampled using different downsampling rates. The downsampling rate means the proportion of the sampled cells. For each downsampling rate, we evaluated clustering results based on the average (over the scRNA and scATAC samples) error rate in cluster assignment (see Table R1 below). These results showed that DC3 performed well in the joint clustering tasks using a small number of cells (178 cells, using only 20% cells). However, when the cell numbers became too small, DC3 was not able to perform joint clustering well.

We also evaluated deconvolution performance by the mean Pearson correlation coefficient (PCC) between the subpopulation-specific loop profiles inferred by DC3 and the corresponding observed loop profiles in the two cell lines under different downsampling rates in Table R6. These results demonstrated DC3 maintained strong deconvolution performance when we downsampled the cells using 20% downsampling rate. However, similar to the joint clustering performance, DC3 was not able to deconvolute the bulk HiChIP when the cell numbers became too small. In this version, we have added the downsampling cell experiments in our Results section.

Table R1. Performance of DC3 under different downsampling rates (50 runs).

Downsampling rate	100%	50%	20%	10%
Joint clustering performance (error rate)	0/892	0.52/446	10.23/178	25.76/89
deconvolution performance in GM1278 (PCC)	0.93	0.87	0.78	0.57
deconvolution performance in K562 (PCC)	0.81	0.79	0.68	0.50

Comment: *Is the result robust to the number of factors (in nmf) chosen?*

Answer: We calculated clustering stability based on the method in Brunet *et al.* (reference 35 in paper) for K ranging from 2 to 5 (see Supplementary Figure 23). The results are consistent with our previous work that clustering results for RA day 4 are stable when $K = 2$ or 3, and the results are not stable when K is increased to 4 or 5. Hence, we set $K = 3$ for the remaining of the analysis. We have added the clustering stability results to “Methods: Initialization and parameters selection” and Supplementary Figure 23.

Comment: *Typos:*

Line 150: “We now want [to] assess”

Line 152: “We focused on one [of] the subpopulation[s]”

Line 159: “we performed [a] HiChIP”

Response: Thanks you for catching these typos. We have corrected them in this revision.

Reviewers' comments:

Reviewer #1 (Remarks to the Author):

The manuscript has improved relative to the previous version. Some additional clarifications are needed, as outlined below.

Replace the sentence "The interpretation of the NMF terms was already given in Duren et al." with a brief summary of the interpretation, including a cite to Duren et al.

This sentence is confusing: "Although the optimization can provide initial estimates of subpopulation-specific profiles (subpopulation profiles), in simulation experiments we observed that when single cell data is available, it is always better to estimate a subpopulation profile by averaging the data from the single cells assigned to that subpopulation (Methods)." I think the problem arises in part from the word "initial." In what sense does the optimization provide initial estimates? It seems that initial estimates would need to be provided prior to the estimation. If the estimates are a product of the optimization, then the text should say specifically what component of the model is being treated as an estimate in this case. (I guess they come from W_1 and W_2). Then evidence needs to be given to support the claim that "it is always better to estimate a subpopulation profile by averaging the data from the single cells assigned to that subpopulation." I guess this claim must be based on empirical evidence, in which case the word "always" is not appropriate.

I could not find this: "we have underlined the definition of candidate enhancer sets in Results: The DC3 algorithm. In addition, we also added description of the hyperparameters which can be tuned automatically in the same section." Note that the Methods section seems to use the word "parameter" to refer to hyperparameters. This should be consistent.

The information in the reviewer response document about how GO terms are selected should be added to Methods.

Reviewer #2 (Remarks to the Author):

The authors have addressed my comments.

Point-by-point responses to Reviewer 1

Comment: Replace the sentence “The interpretation of the NMF terms was already given in Duren et al.” with a brief summary of the interpretation, including a cite to Duren et al.

Response: Following your suggestion, we have revised the description of the interpretation of the NMF terms as “As previously described⁶, each NMF term drives the decomposition of a single-cell data matrix into two factors W and H , with columns of W representing cluster-specific profiles, and each column of H giving the relative weights (for cluster-assignment) of a particular single cell.”

Comment: This sentence is confusing: “Although the optimization can provide initial estimates of subpopulation-specific profiles (subpopulation profiles), in simulation experiments we observed that when single cell data is available, it is always better to estimate a subpopulation profile by averaging the data from the single cells assigned to that subpopulation (Methods).” I think the problem arises in part from the word “initial”. In what sense does the optimization provide initial estimates? It seems that initial estimates would need to be provided prior to the estimation. If the estimates are a product of the optimization, then the text should say specifically what component of the model is being treated as an estimate in this case. (I guess they come from W_1 and W_2). Then evidence needs to be given to support the claim that “it is always better to estimate a subpopulation profile by averaging the data from the single cells assigned to that subpopulation.” I guess this claim must be based on empirical evidence, in which case the word “always” is not appropriate.

Response: Following your suggestion, we have removed the word “initial” and explained that the estimates of the subpopulation profiles can be obtained from W matrix in the NMF term.

Also, we have removed the word “always” and provided empirical evidence (in Supplementary Table 1) for the superior performance of the mean single cell profiles over the profiles obtained from the W matrix.

Comment: I could not find this: “We have underlined the definition of candidate enhancer sets in Results: The DC3 algorithm. In addition, we also added description of the hyperparameters which can be tuned automatically in the same section.” Note that the Methods section seems to use the word “parameters” to refer to hyperparameters. This should be consistent.

Response: Thank you for raising these points. We have added the following definition of candidate enhancer sets in the description of the DC3 algorithm in Results: “Note that, instead of using a pre-defining enhancer set, DC3 defines the candidate enhancers directly based on ATAC-seq and HiChIP data”. We have also changed “parameters” in the Methods section to “hyperparameters”.

Comment: The information in the reviewer response document about how GO terms are selected should be added to Methods.

Response: Following your suggestion, we have added the detail description in “Methods: GO terms selection”.

REVIEWERS' COMMENTS:

Reviewer #1 (Remarks to the Author):

The authors have addressed all of my remaining concerns.

Point-by-point responses to Reviewer 1

***Comment:** The authors have addressed all of my remaining concerns.*

Response: Thank you for reviewing our manuscript. We are also very grateful for your detailed comments and helpful suggestions in the review process.